# Three-Dimensional Printing of Foods: A Critical Review of the Present State in Healthcare Applications, and Potential Risks and Benefits

**DOI:** 10.3390/foods12173287

**Published:** 2023-09-01

**Authors:** Wenxi Zhu, Michèle M. Iskandar, Vahid Baeghbali, Stan Kubow

**Affiliations:** 1School of Human Nutrition, McGill University, Montreal, QC H9X 3V9, Canada; wenxi.zhu@mail.mcgill.ca (W.Z.); michele.iskandar@mcgill.ca (M.M.I.); 2Food and Markets Department, Natural Resources Institute, University of Greenwich, Medway, Kent ME4 4TB, UK; v.baeghbali@greenwich.ac.uk

**Keywords:** 3D printing, risks and benefits, food quality, acceptability

## Abstract

Three-dimensional printing is one of the most precise manufacturing technologies with a wide variety of applications. Three-dimensional food printing offers potential benefits for food production in terms of modifying texture, personalized nutrition, and adaptation to specific consumers’ needs, among others. It could enable innovative and complex foods to be presented attractively, create uniquely textured foods tailored to patients with dysphagia, and support sustainability by reducing waste, utilizing by-products, and incorporating eco-friendly ingredients. Notable applications to date include, but are not limited to, printing novel shapes and complex geometries from candy, chocolate, or pasta, and bio-printed meats. The main challenges of 3D printing include nutritional quality and manufacturing issues. Currently, little research has explored the impact of 3D food printing on nutrient density, bioaccessibility/bioavailability, and the impact of matrix integrity loss on diet quality. The technology also faces challenges such as consumer acceptability, food safety and regulatory concerns. Possible adverse health effects due to overconsumption or the ultra-processed nature of 3D printed foods are major potential pitfalls. This review describes the state-of-the-art of 3D food printing technology from a nutritional perspective, highlighting potential applications and current limitations of this technology, and discusses the potential nutritional risks and benefits of 3D food printing.

## 1. Introduction

Additive manufacturing, also known as three-dimensional printing (3D printing), is a powerful tool for producing cost-effective devices based on layer-by-layer deposition technology [1]. Nowadays, 3D printing technology is used in multiple industries such as architecture and construction [2], aerospace and automotive [3], the energy industry [4], and medical industry [5]. Three-dimensional printing technology has also shown its potential for applications in the food and nutrition context [6,7]. Because of its digital design, accurate quality control, environmentally friendly, energy-efficient, and low-cost characteristics, 3D food printing plays a beneficial role in food manufacturing. The advantages of using 3D food printing in manufacturing include customizing the shape and content of food, enhancing the utilization of ingredients, and creating food that meets personalized nutritional needs [6,8,9]. Current applications include chocolate [10], meat and meat alternatives [11,12], egg white protein objects [8], and fruit and vegetable smoothies [13]. Several restaurants have used 3D food printers to provide customers with visually appealing food and unique dining experiences [14,15]. Three-dimensional printed food is also used to support sustainability. For instance, unshaped fruits and vegetables or insect proteins were used to create 3D printed food for reducing food waste and CO_2_ emissions from livestock [14,16,17].

Four main forms of 3D printing of food products have been introduced, which include extrusion, inkjet printing, binder jetting, and powder bed melting through selective laser sintering [18]. All forms are dependent on the use of 3D computer-aided design [9]. An extrusion method is a common approach in 3D food printing [19]. The extrusion technique allows fresh ingredients, or pre-processed ingredients, to be printed layer by layer and deposited onto a platform until the designed 3D structure is shaped [6,7]. Inkjet printing is used for low-viscosity materials, so it is primarily used for 2D printing, such as creating high-resolution images on cookies [20,21]. Binder jetting and selective laser sintering work primarily with powder-based materials, where the powder particles combine in the presence of heat or liquid as a binding agent [19].

Several studies have investigated food printing technology. Researchers believe that potential applications for 3D printing could play a beneficial role in food and nutrition, including texture-modified diets and personalized nutrition development, as a possible solution to food insecurity, and a way to introduce new ingredients into the diet [1,13,22,23]. Three-dimensional food printing may improve the appearance of soft or pureed textured foods, which can help individuals with swallowing difficulties and prevent aspiration pneumonia caused by choking, as well as enhance patients’ appetite and improve malnutrition [23,24]. This novel technology also allows the creation of personalized foods based on individual nutritional needs, including customized individual supplements, customized patient-oriented diets, and personalized probiotics and nutrients enclosed into functional foods through microencapsulation technology [6,18,22,25]. Incorporating new ingredients into conventional foods is also one of the promises of 3D food printing, which replicates the appearance of foods that have nutritional value but are not acceptable to consumers to increase acceptability and consumption [11,17,26]. In addition, this technology may reduce regional food insecurity by increasing sustainable food sources, reducing food waste, and improving diet quality [27,28].

Currently, most research on 3D printed food focuses on the optimization of 3D printing technology, the properties and composition of food materials, or consumer attitudes towards 3D printed food [8,29,30,31]. Several reviews have summarized the potential benefits of 3D printed foods in the food and nutrition sector [9,32]. On the contrary, to date, no large-scale clinical trials focusing on the effects of 3D printed food intake on patients’ nutritional status or diet quality have been published [33]. Few studies have been conducted to discuss the challenges and risks of 3D printed foods in the health sector [34]. Emerging 3D food printing technology offers the opportunity to customize foods, dysphagia diets, and personalized nutrition, but the effects of the 3D printing process on nutrient retention have not been previously studied. In particular, the 3D printing technology may disrupt the integrity of the food matrix, the bioaccessibility of micronutrients through ultra-processing, extrusion, or high temperatures, and subsequently affect the nutritional quality of the final product. Therefore, this narrative review focuses on the research of 3D printed foods in nutrition, including current studies and applications, potential benefits, and highlighting the limitations and potential pitfalls of 3D printed foods in healthcare applications.

## 2. Promises

### 2.1. Nutritional Therapy

Visually appealing modified-texture diet development has become one of the applications of 3D printing in nutritional therapy [23]. Modified texture diets, such as thickened liquids and purees, could help patients with swallowing difficulties caused by stroke, head and neck cancer, neuromuscular disorders, to prevent aspiration pneumonia and enhance diet quality [35,36,37,38]. Patients with swallowing difficulties often have inadequate calorie intake. They are at a higher risk of malnutrition because texture modification involves a step of dilution with water or other liquids in the preparation of the mixture [23,39]. Three-dimensional printing may use fortified liquids to create food inks, such as protein-rich liquids to increase nutritional density [34]. Three-dimensional printed foods may allow adding microgels and latex gels as thickeners for modified-texture foods. These gels can not only create proper texture and increase nutritional density for printed foods, but also add dietary fiber or bioactive compounds such as capsaicinoids and β-carotene to the gel to boost the nutritional value of the product [23,40,41]. Antioxidants can also be inserted into food through 3D printing and microencapsulation technology [42]. Future trends to enhance the nutritional value of dysphagia diets may involve the use of microencapsulation technology in combination with 3D printing. Among the benefits, microencapsulation allows the incorporation of bioactive hydrophobic substances such as tocopherols and polyphenols with low water solubility in food 3D printing technology, which can circumvent issues such as poor bioavailability and sensitivities to degradation due to temperature, pH, and oxidation [43].

In addition, the traditional pureed dysphagia diet disrupts the appearance of prototypical foods, which may reduce the patient’s appetite leading to reduced food intake and increase risk of malnutrition [33,44]. In contrast to handmade shaping and silicone molds, 3D printed foods can create attractive appearances of modified-texture food faster, and natural colors can be incorporated to offset the color lost in food preparation [23]. Kouzani et al. [37] designed a 3D printed dysphagia diet using tuna, pumpkin puree, and beetroot puree, and they concluded that 3D printing technology reduced the time to make the dysphagia diet, improved product consistency, and enhanced the acceptability of texture-modified foods. Classification of dysphagia diets and careful consideration of food textures are necessary to provide the most appropriate foods for patients with dysphagia, which contributes to their safety and health [45]. According to the framework of the International Dysphagia Diet Standardization Initiative (IDDSI), a global initiative recommending the implementation of standardized terminology worldwide, dysphagia diets can be classified into eight levels (0 to 7) [46,47]. Three-dimensional food printing technology allows more accurate standardization of printed food textures through computer control and digital design, which will meet the hardness, adhesion, and cohesiveness requirement for each class in the framework [6,18,23]. In addition to dysphagia diets, modified-texture food by 3D printing can be applied to children’s food. Some studies have shown that children are less likely to prefer hard foods and foods with particles than adults [48,49]. The texture of food affects a child’s food preferences and may lead to picky eating in children [48]. Three-dimensional foods may be able to create more child-friendly foods by adjusting the texture of the food. Children’s preference for vegetables was found to be low compared to other core food groups such as dairy, meat and cereals [50]. By printing food, appealing shapes and textures of vegetables can be created to encourage children to consume more vegetables [6,51].

### 2.2. Personalized Nutrition

Another important application of 3D food printing technology in the field of nutrition is personalized nutrition [9,15,19]. Three-dimensional printing technology can be designed to produce foods that meet individual nutritional needs based on their nutritional status, lifestyle, and dietary preferences [52,53]. Food printers may be able to help consumers achieve a well-balanced diet through nutritional customization, where customers control their diet and accurately count calories by selecting the number and type of ingredients and the corresponding production parameters through the printer operator interface [6]. Personalized nutrition and precision nutrition are now being used as strategies to prevent non-communicable chronic diseases [54]. Three-dimensional food printing technology allows for the adjustment of macronutrients and micronutrient concentrations on food models to satisfy the demand of people with specific food-related diseases. For example, foods or diets could be printed to reduce sodium and potassium content for patients with chronic kidney disease and introduce high dietary fiber for obese patients [14,34]. Three-dimensional printing technology can also fortify food inks with nutrients such as vitamins, minerals, and probiotics to create personalized foods for athletes, pregnant women, military personnel, and children to meet the nutritional needs of special populations [14,52,55,56]. In a military environment, soldiers’ physiology can be detected by real-time sensors, and the physiological signs data can be transmitted back to 3D food printers to create food that meets the soldiers’ individual needs to improve their performance [14,52]. Derossi et al. [55] had customized a children’s fruit snack through 3D printing technology which contained calcium, iron, and vitamin D to meet the nutritional demands of children between the ages of 3 and 10. In addition to customization of nutrition, 3D printing technology allows consumers the freedom to create personalized food shapes and flavors [7,57]. Consumers can participate in the design and printing of foods, which may have a positive impact on consumer satisfaction [52]. Sun et al. [6] reported that a workshop of 3D printed cookies attracted many children to watch the printing process and taste the cookies. According to Burke-Shyne et al. [34], the appealing shape of 3D printed vegetables may be able to boost children’s vegetable intake. Nutritional customization and fortification of food by 3D food printing may be a potential way to address nutrient deficiencies. When 3D food printers are widely used, 3D printed food may be able to alleviate the food insecurity caused by invisible hunger in developing countries [14,28,34].

### 2.3. Support of Sustainability

Another prospect of 3D food printing is to support sustainability [16,58]. Because 3D printing allows for food customization, consumers may be more likely to eat all the food that meets their personal preferences, reducing food waste [14,58]. Three-dimensional food printing may be able to enhance food utilization and reduce food waste by using deformed vegetables and fruits that do not meet the criteria for sale, vegetable and fruit or animal by-products to make more palatable food products for consumers [26,58,59,60]. Feng et al. [26] investigated the possibility of using ground potato by-products mixed with yam powder to make air-fried snacks. Three-dimensional food printing offers the possibility of introducing alternative protein sources from plant-based or cultured meat to algae and insects, which are not frequently consumed or well-accepted in many western societies, in desirable shapes and tastes. Using alternative protein sources may address nutritional aspects of the food security global challenge [17,28,60,61]. For example, edible insects are not only rich in nutrients, containing high-quality proteins, vitamins, and amino acids, but also have a lower carbon footprint [62]. Severini et al. [17] incorporated yellow mealworm powder into cereal snacks through 3D printing technology, and the results showed an increase in protein content. Therefore, provided edible insects are widely accepted by consumers, using insect powders in food formulations may be a feasible solution to increase the nutrition density of food products and address undernourishment in the least developed countries. In addition to 3D printed meat alternatives, 3D food printing technology is also being developed for cultured meat in order to reduce carbon emissions and environmental pollution [12]. In cultured meat production, the risk of animal-borne diseases and food-borne illnesses is lower because there may be no need to handle livestock [63]. Three-dimensional food printing allows the use of long shelf-life capsulated food inks [16,64]. Therefore, 3D printed food technology can reduce food waste and post-harvest losses through the capsulation of ingredients compared to fresh ingredients [16]. Three-dimensional food printing can also impact food availability not only using alternative ingredients and resource utilization, but also using long shelf-life capsules. Those capsules can be used to print fresh food even in remote areas and “difficult” situations, such as military and space missions, and ultimately in humanitarian emergencies [16,65]. Table 1 presents a summary of studies on the printability, consumer acceptability, and/or applicability of 3D printing of novel foods and functional products.

## 3. Current Applications

### 3.1. Functional Foods

There has been a lot of research into the formulation and optimization of 3D printed foods. However, only a few research groups are actively involved in the study of printable formulations suitable for the creation of 3D functional foods, and studies of 3D printed foods directed at achieving nutritional goals are still limited [14]. Derossi et al. [55] modified a fruit-based puree consisting of a banana, dried non-fat milk, dried mushrooms with pectin into a food ink to create children’s snacks which met the calcium, iron, and vitamin D intake requirements of children from 3 to 10 years old. Lille et al. [60] utilized 3D printing technology to produce healthy structures rich in fiber and protein but low in fat or sugar. Cereal snacks containing probiotics have also been made using 3D printing technology to print a “honeycomb” structure to reduce the inactivation of probiotics after baking [56].

### 3.2. Three-Dimensional Printing Manufacturers

The global 3D food printing market is expected to grow from USD 98.71 Million in 2019 to USD 472.95 Million by the end of 2025 [66]. There are several 3D food printers on the market today, including those designed for industrial production and those for use in homes or restaurants [19]. They are available in the price range of $300–$6000 [14]. Three-dimensional printers combine hardware and software to print food. Most food printers are equipped with a user-friendly interface and pre-loaded recipes, or specific food capsules and materials. Three-dimensional printing designs can be computer-based or carried out directly on the interface of some 3D printers [67]. There are some examples of commercially available 3D food printers. The Focus (By Flow, Eindhoven, The Netherlands) is a 3D printer for the manufacturing of pasty foods, which allows users to print out meals by downloading recipes (Figure 1a) [68]. Choc Creator V2 Plus (The Choc Edge, Exeter, UK) is a food printer specially designed for printing 3D chocolate (Figure 1b) [69]. This machine can be used in the restaurant or hospitality industry to make artistic chocolates for customers. Foodini (Natural Machines, Barcelona, Spain) is a small printer for professional and domestic use (Figure 1c). It works with an open capsule system where the user can fill in fresh ingredients to print the designed paste-like foods [64]. The makers of the Foodini food printer have partnered with a restaurant in Barcelona, Spain, to experiment with it in preparing food [64]. The Createbot 3D Food printer (Createbot, Ningbo, China) is a multi-material food 3D printer, and can print a variety of paste substances, including cookie batter, mashed potatoes, and chocolate (Figure 1d) [70]. ORD Solutions RoVaPaste (ORD Solutions, Cambridge, ON, Canada) is a multi-material printer for home and catering able to print 3D with materials other than food, such as clay or silicone [71].

### 3.3. Novel 3D Food Products

Three-dimensional printing has not yet been put into industrial production on a large-scale, because of printing speed and material texture restrictions [12,18,57]. Currently, a limited number of commercial food products are available through 3D printing technology. A pasta company, Barilla (Parma, Italy), has collaborated with the Netherlands Organization for Applied Scientific Research (TNO) to develop research on 3D pasta printers, which can print unusual shapes of pasta from dough made of pure durum wheat semolina and water. Customers can choose personalized pasta patterns and textures on the company’s e-commerce platform, and prices for the product range from 24.90 € to 55.67 €. However, the pasta is not yet available on the market [72]. The vitamin supplement company Nourished (Bristol, UK) is using 3D printing technology to design and produce multivitamin gummies that meet individual nutritional requirements. Consumers can choose the seven micronutrients they want to add to their gummies based on a questionnaire or personal preference on the website. The product became available in 2020 [73,74]. La MIAM Factory (Namur, Belgium) is a confectionery factory dedicated to 3D chocolate printing. The company can print chocolates in various shapes according to the customer’s design, and current prices are from €3.50 to €12.50 depending on the size of the chocolate [75]. Poseidn is a Canadian food company that uses 3D printing technology to create solid drinks [76]. The beverage powder is dried, and 3D printed into various animal, flower and even game console shapes. Some nutrients are added to these solid drinks as functional foods. The product is currently $2.99 CAD each for the hot beverage series, while the functional beverage series is planned to become available later [76]. There are also several food companies on the market working to create cultured meat or meat alternatives using 3D printing technology. Novameat (Barcelona, Spain) prints vegan meat using ingredients such as peas, seaweed, and rice that provide a meat-like texture and nutritional value [77]. Another company, SavorEat (Rehovot, Israel), through proprietary 3D printing technology, automated cooking devices to produce veggie burger meat, in which cellulose derivatives are used as binders to simulate the texture of meat. They plan to develop sustainable agriculture through industrial printing of meat and meat alternatives and avoiding slaughter [78]. However, although the 3D printing industry and market are expected to grow substantially over the next few years [66], techno-economic feasibility assessments in various settings such as healthcare institutions, homes, or the industry, are lacking [79]. In terms of scale of production, it has been observed that presently, most existing 3D printing systems are not devised for large-scale uses. Furthermore, it appears that there is still a shortage of clearly defined customer segments and effective profit strategies or mechanisms [80].

## 4. Challenges

### 4.1. Consumer Acceptance

The attitudes of consumers and nutrition-related professionals toward 3D printed foods are critical to enabling 3D printed (3DP) foods to achieve their promise in the nutrition sector. The promise of 3D printing to produce dysphagia diets and incorporate nutrients will not be realized if patients are unable to accept 3D printed foods. Brunner, Delley [29] surveyed 260 German-speaking adult residents from Switzerland about their attitudes towards 3DP foods via a questionnaire sent by mail, including the willingness to consume, considerations of health and natural ingredients, and aversion to novelty. The survey results show that these potential consumers have negative attitudes towards 3D printed foods, mainly due to a lack of acceptance of new foods and new food technologies. While knowledge of the benefits and potential applications of 3D printed foods has increased consumer willingness, these consumers may still be very cautious about the technology. A four-day online group interview with 30 Australian participants collected consumer attitudes towards 3D printed food, including the printed meat and insect protein incorporation [81]. Because participants were completely unfamiliar with 3D printed foods before the experiment, they were cautious about 3D printed foods. They indicated low acceptability even if they had an initial understanding of the possible health benefits of 3D printed foods. Participants disagreed that 3DP foods would be better than authentic foods because they are “unnatural” and “ultra-processed”. Nonetheless, a study of consumer attitudes toward 3D printed foods in a military setting showed a reduction in food and food technophobia after volunteers consumed real 3D printed energy bars [52]. Currently, consumers are cautious about 3D printed food, and there is still a need to increase consumer understanding of 3D printed food and alleviate their phobia of new foods and technologies. Some professionals with nutrition backgrounds are also uncertain about the development of 3D printed food in the food and nutrition sector [34,82]. The concerns of food professionals and scientists about 3D printed food may limit the application of 3D printing in this field. Burke-Shyne et al. [34] interviewed ten experts with experience in operating 3D food printers, including five nutrition experts. Interviewees recognized the potential role of 3D food printing in the field of nutrition, but believed that 3D printed food faces challenges such as food safety issues and nutritional accessibility due to changes in the nutritional matrix of food [34]. Another group interviewed on the 3D food printer as a teaching tool in nutrition education collected the attitudes of nutrition students and dietitians towards 3D printed food and 3D food printers [82]. The results show that students and teachers from nutrition backgrounds are concerned about the safety of 3D printed food and believe that 3D printed food might confuse and frighten consumers. At this stage, 3D food printers may not be introduced into the classroom as a teaching tool [82]. In addition to 3D food printing, 3D printing-based personalized nutrition also faces the challenge of consumer acceptability. At this stage, personalized nutrition is still a new concept. Frewer et al. [83] proposed that when consumers do not understand the concept of new technology, they do not tend to try to adopt it. Their team compared consumer attitudes toward seven new food-related technologies, including genetically modified foods, personalized nutrition, and nanotechnology. Because personalized nutrition may involve genetic testing and nutrigenomics, consumers may be hesitant to embrace it due to ethical, social, and private disclosure concerns [83]. de Roos [84] argued that low consumer acceptance, privacy concerns, and the lack of regulations limit the commercialization of personalized nutrition. Concerns about 3D printed foods and personalized nutrition may be a challenge to achieve personalized functional foods based on 3D food printing technology.

### 4.2. Nutritional Issues

The nutritional quality of 3D printed food is also one of the challenges of whether 3D printing can be applied in the field of nutrition. In the studies of the attitudes of consumers and professionals with a nutrition and food background toward 3D printed food, interviewees were concerned about the nutritional quality of 3D printed food and believed that 3D food printing technology may destroy the original nutritional value of the food [29,34,81,82]. The processes involved in 3D food printing and the post-processing (steaming or baking) of 3D printed food may affect nutritional quality [6,85]. Many chemical reactions and physical changes occur during processing and post-processing (such as protein denaturation, starch gelatinization and fragmentation, and moisture reduction) that may change the nutritional value of the final product [6,85,86]. Martínez-Monzó et al. [86] compared the concentration of vitamin E acetate in peanut butter before, after 3D printing, and after 3D printing and thermal post-processing, and showed that the extrusion process of 3D printing contributed to a slight decrease in vitamin E acetate, but the thermal treatment led to a significant decrease in the concentration of vitamin E acetate. Sun et al. [6] discussed the differences between food extrusion cooking and extrusion-based food printing, and argued that 3D printing-based food extrusion is digitally controlled in traditional food extrusion processing, but the process parameters of extrusion-based 3D food printing include similar parameters related to traditional food extrusion cooking, such as temperature, shear force, and pressure. The high temperature, high pressure, and high shear characteristics of extrusion cooking caused biochemical reactions such as protein denaturation, starch gelatinization, lipid modifications, microbial and enzyme inactivation, the formation of volatile flavor components, and increased insoluble dietary fiber [87]. Extrusion cooking also has the potential to improve the nutritional quality of the product by improving the digestibility of starch and protein and increasing the retention rate of bioactive substances with antioxidant properties [87]. During extrusion, the starch granules become smaller and produce more digestible fragments, increasing digestible carbohydrate content [88]. The high temperature and dehydration in extrusion cooking can lead to a Maillard reaction, and excessive Maillard browning can lead to loss of lysine, destruction of vitamins, and decreased bioavailability of other micronutrients [87]. Extrusion may result in the destruction or alteration of natural bioactive compounds [89]. Altan et al. [90] investigated the effect of extrusion on antioxidant activity, total phenolic content and β-glucan content in a barley flour-based food model. The result of this study showed a 60–68% reduction in antioxidant capacity and 46–60% reduction in total phenolic content in barley extrudates compared to unprocessed barley flour. In 3D food printing, selective laser sintering, hot air sintering, and extrusion (hot-melt extrusion) may include thermal processing [19]. Thermal processing may reduce the stability of nutrients, for example, thermal processing of fruits and vegetables has shown significant losses of vitamin C and thiamin [91]. Vitamin C concentrations measured immediately after thermal processing were 15% to 45% of those of fresh products; thiamin was reduced by about 50% during heat processing [91]. However, no research team has yet studied the effect of thermal processes on nutrient bioavailability during 3D food printing. Currently, few researchers have investigated the effect of specific 3D printing technologies on the nutritional value of printed foods, and more research is needed on the effect of other parameters in 3D food printing technologies on the nutritional value of the final product.

### 4.3. Food Safety

The technical issues of 3D food printing and food safety on 3D printed food have been mentioned in many review studies [9,14,18,34]. Since most 3D printed food products are pastes, the shelf life may be limited. For example, the structural rheology of dough or dough prepared for 3D printing often changes after 2 h of production [92]. In long-term care centers or hospitals, food made with 3D printing may need to sit for a period of time before being distributed to patients [34]. The instability of 3D printed foods may limit the potential applications of 3D food printing in creating appealing difficult-to-swallow foods. Most 3D food printing processes involve heating during the extrusion process and cooling after deposition [57]. In the clinical setting, 3D printed meals also face the challenge of reheating [34]. These heating and cooling or reheating processes may make it easier for microorganisms to grow and affect food safety [57]. In addition, there is direct contact between the components of the 3D printer (e.g., nozzles, trays) and the raw materials produced during the food preparation process. In a study of printed fruit and vegetable smoothies, Severini et al. [13] found initial psychrophilic, mesophilic, and yeast levels of 4.27, 5.02, and 4.23 log CFU/g, respectively. These high values implied that contact between food ink and printer components during the 3D printing process increases microbial contamination [13]. Sun et al. [6] mentioned that most domestic desktop printers are made of plastic and may release ultrafine particles. Such toxic particles can be released during the printing process and cause adverse health effects. Most researchers believe that the safety issues of 3D printed food will be overcome in the future [34,57]. However, present concerns about the safety of 3D printed food may slow down the development of 3D printed food applications in the nutritional field. There are no national, provincial, or international mandatory standards specifically for 3D printed foods, which could lead to confusion regarding them. Inconsistent standards for 3D food printing could result in 3D printed foods not being used in nutritional medical therapy. According to Health Canada [93] 3DP foods may be under the category of “novel foods”. With no history of safe use, 3D printed foods will need to be approved by Health Canada before they can be applied in the nutrition and food sector. Figure 2 shows a summary of the challenges of feasible applications of 3D food printing technology in the food and healthcare sectors.

## 5. Pitfalls

### 5.1. Health Risks of Ultraprocessed Foods

Three-dimensional food printing may involve health risks. Currently, the Nova system is the most-used way to check the classification of foods and diets based on the degree of food processing [94]. Nova categorizes all foods into four groups according to the nature, degree, and purpose of their processing: unprocessed or minimally processed foods, processed cooking ingredients, processed foods, and ultra-processed foods [94]. Since 3D printed foods require pre-processing, post-processing of food materials and processing in the 3D printer, as well as food additives that may be necessarily incorporated, they can be classified as ultra-processed foods [95,96]. In interviews with professionals with nutritional backgrounds, interviewees also considered 3D printed foods to be ultra-processed foods and pointed out that they may have some adverse health effects [34,82]. Current studies consistently show that ultra-processed foods are nutritionally unbalanced and that their consumption negatively affects the nutritional quality of the diet [97,98,99]. Fardet [100] collected the data of 98 ready-to-eat foods and concluded that ultra-processed foods lead to a stronger glycemic response and lower satiety compared to raw and minimally processed foods, which could result in metabolic symptoms. A Cohort study investigated the association between ultra-processed food consumption and functional gastrointestinal disorders, and they found that an increased proportion of ultra-processed food in the diet was associated with a higher prevalence of irritable bowel syndrome [101]. Consumption of 3D printed foods as ultra-processed foods may reduce the quality of diets and lead to similar adverse health outcomes. Because of the potential benefits of home food printers (quick cooking, meeting personal preference, special design, etc.), they may be popular with consumers in the future and be as widely used as microwave ovens [6,102]. The widespread consumption of 3D printed food as a convenience food or ultra-processed food may reduce diet quality [103]. Concerns about 3D printed foods also include rapid preparation similar to microwave-like foods; low satiety due to ultra-processing, mindless eating that may result from designs that satisfy personal preferences, all factors that induce passive energy overconsumption [94]. For example, a two-week in-patient intervention trial showed that an ultra-processed diet resulted in weight gain and enhanced caloric intake by approximately 500 kcal/day. In contrast, an unprocessed diet given to the same subjects containing similar amounts of calories, macronutrients, fiber, sugars, sodium, and fiber resulted in spontaneous weight loss [104]. A variety of mechanisms of action have been postulated for such findings, which includes that over-consumption of calories is reinforced from the intake of hyper-palatable, ultra-processed foods high in sugar and fat [105].

### 5.2. Loss of Food Matrix

Pretreatment technologies (crushing, gelation) of 3D printed food material may change the food structure and lose the food matrix [34,95]. Losing the natural matrix or microstructure resulting from processing may affect the release, transformation, and subsequent absorption of certain nutrients in the gastrointestinal tract [106]. Loss of the natural food matrix has been indicated to lead to altered patterns of intestinal digestion and absorption of food components. Nutrients from ultra-processed foods may be absorbed in more proximal regions of the gut, which can lead to differing patterns of gut–brain signals from those generated from the same nutrients from whole food intake [103]. Signals affecting food intake and digestion, energy balance and body weight come from enteroendocrine hormones (i.e., ghrelin, cholecystokinin, glucagon-like peptide 1, peptide YY, insulin-like peptide 5), vagal afferent nerves innervating the gastrointestinal tract (the neuronal component of the gut–brain axis) and microbiota-derived metabolites (i.e., short-chain fatty acids, amino acids, bile acids, and γ-aminobutyric acid) [107]. Some studies mentioned that losing food matrices affected the bioavailability of polyphenols and folic acid, the digestibility of proteins and carbohydrates, potentially impacting on glycaemia control, satiety, and gut health [106,108]. Therefore, changing the food matrix in 3D food printing may affect the nutritional value of the final product. Vucea et al. [109] investigated the use of modified-texture foods and the nutritional status of the elderly in 32 long-term care facilities in four Canadian provinces. The results showed that even with a texture modified diet, consumers still had lower intakes of energy, protein, and several micronutrients. The study has suggested that pureed food contains lower amounts of energy, protein and micronutrients compared to the regular texture [110]. Current concern about texture-modified meals made by 3D printing is that long-term intake of 3D printed meals may result in the loss of certain food components associated with whole foods, which could result in suboptimal nutritional status.

Most 3D printed food products are paste-like in texture and require the addition of food-grade additives used to improve the viscosity and printability of the material [19]. Many studies have investigated the addition of hydrocolloids to enhance the printability of 3D printed food inks [55,111]. Although hydrocolloids have health benefits in terms of increasing satiety, limiting lipid digestion and slowing starch digestion [8,13,55,96,111], the use of other food additives (sweeteners, emulsifiers) in 3D printed food may affect the human gut microbiota and exert adverse health effects [112,113]. For example, the obesogenic effects of food additives such as emulsifying agents demonstrated in animal models have been related to altered gut microbiota profiles as seen in germ-free mouse and fecal transplant studies [114].

### 5.3. Disruption of Food Cooking Traditions

When 3D printed food is widely applied in the household, it may challenge a long-standing cultural heritage of family food preparation and meal eating. Burke-Shyne et al., [34] mentioned that some traditional cultures regard cooking and cooking as a social activity that brings friends and family together and emphasizes the appreciation of the ingredients and the process of food preparation. Three-dimensional food printing may diminish the social properties of cooking. Eating together with others has been defined as an important daily practice [115]. Studies show that eating with others can maintain good eating habits and better cope with stress [115,116]. A Japanese gerontological study collected body weight and diet status of elderly adults living alone through a questionnaire and showed that eating alone may lead to obesity or being underweight [116].

## 6. Discussion and Conclusions

Three-dimensional printing is a promising novel technology in the field of food and nutrition. Three-dimensional printed food has potential applications in the field of nutrition, including the production of visually appealing modified-texture diets, personalized nutrition, supporting sustainable development and mitigating food insecurity. Three-dimensional food printing is fast and easy, and operators can produce printed food by simply learning parameter settings or downloading electronic recipes directly. Many 3D food printers are already available for professional, industrial, and domestic use, and in the future may become suitable for producing ready-to-eat foods that meet individual nutritional demand through devices such as vending machines connected to personal workout APPs. The realization of the above applications in the field of 3D food printing for nutrition still faces many challenges, including the stability of 3D printed food, gaining consumer trust, and a better understanding of the nutritional impact of printed food on nutrient density and bioavailability. Once these challenges are overcome, 3D food printing technology can be more widely used. However, once 3D printed foods are accepted and widely used by consumers, their overconsumption may cause adverse health consequences due to ultra-processing, usage of food additives, and destruction of the food matrix. Three-dimensional printed food may also affect present dietary traditions, possibly reducing diet quality. Currently, 3D Printing has to shift from a focus on printing snacks to printing more nutritious foods; the emphasis should be placed on improving the nutritional quality of printed foods. More research is needed regarding the nutrient profiles, density, and bioavailability of 3D printed foods, or the health impact of the possible loss of food matrix, such as the effects on the gut microbiome and glycemic response. More clinical research is also needed on the nutritional therapeutic effects of 3D printed texture modified foods on people with swallowing difficulties. Future research should be conducted on the possible applications of 3D printed food in healthcare, such as complementary foods for infants according to developmental stages, supplementary foods for elderly in long-term care, a variety of patient-oriented foods in hospitals, and food products for anorexia, allergies, and intolerances.

## Figures and Tables

**Figure 1 foods-12-03287-f001:**
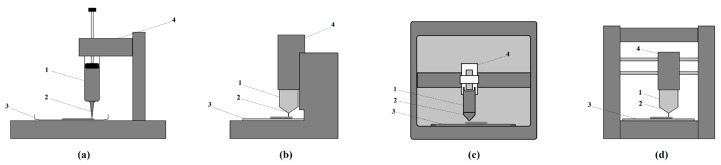
Schematics of some examples of commercially available 3D food printers. (**a**) Focus; (**b**) Choc Creator V2 Plus; (**c**) Foodini; (**d**) Createbot. Where 1 is the extruder assembly, 2 is the nozzle tip, 3 is the printing platform, and 4 is the holder of the extruder assembly.

**Figure 2 foods-12-03287-f002:**
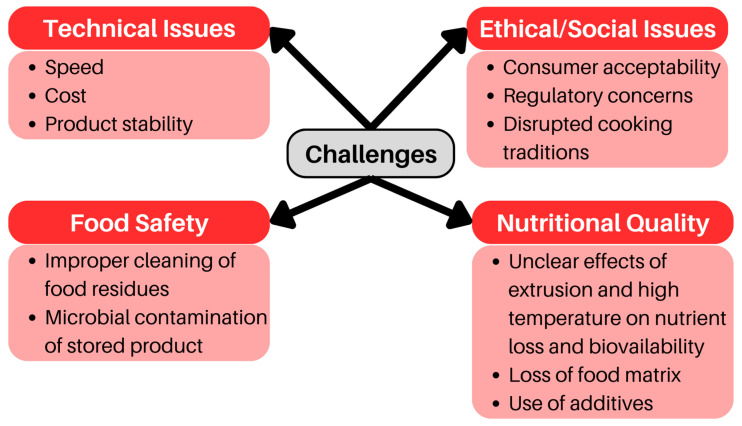
Challenges that need to be overcome to allow widespread applications of 3D food printing technology in healthcare and other sectors.

**Table 1 foods-12-03287-t001:** Overview of studies investigating the printability of novel foods, consumer acceptability and/or applicability to potential functional products.

Objectives	Methodology/3D Printing Technology	Product	Results	Reference
Assess and optimise the printability of snacks containing wheat flour and Yellow mealworm powder	3D Printer mod. Delta 2040 with Clay extruder kit 2.00	3D printed snacks using wheat flour dough with different amounts of ground larvae of yellow mealworms	Addition of mealworm powder affected the printability of the dough, but protein digestibility corrected amino acid score of 3D printed snacks was improved with the addition of insect powder	[17]
Investigate 3D printing characteristics of yam powder with added high-fiber potato by-products	Dual-nozzle 3D printer (SHINNOVE-D1)	Freeze-dried yam and potato processing by-product at varying ratios	Good printing characteristics were obtained for all formulations Addition of potato by-products reduced extrusion swelling and increased WAI and G’	[26]
Establish a 3D printing method for the preparation of 3D food products to be potentially used by people with dysphagia	Envision TEC GmbH Bioploter; pressure controlled extrusion	Tuna print made with butternut pumpkin, beetroot and canned tuna in springwater	Product was found to have similar taste to a similar dish prepared by a skilled cook (3 tasters)	[37]
Preparation of 3D printed functional food with antioxidant properties	Focus 3D food printer with paste printing head and 1.6 mm aperture nozzle	3D printed cookies fortified with freeze-dried microalga Arthrospira platensis encapsulated in alginate microbeads	Cookies were obtained with good shape fidelity with low water activity for high microbiological stability; addition of microalga increased antioxidant activity measured via the ORAC assay but not the ABTS assay	[42]
Survey of sensory evaluation of texture-modified 3D printed chocolate	30 semi-trained panellists evaluated texture and appearance of 3D printed chocolates with varying infill patterns and percentages, using a Da Vinci 2.0 dual nozzle model XYZ printer with ABS filament A survey of consumer perceptions with 244 participants	3 chocolate samples with increasing infill percentages (25, 50, 100%), and a cast commercial chocolate sample as control	Panellists preferred the appearance of the samples with 25 and 50% infill, no difference in preference between the cast and 100% infill samples A generally positive perception of 3D printed chocolates encouraged by prior presentation of 3D printed chocolate samples and a 3D food printer	[51]
Investigate consumer attitudes towards 3D food printing in a military setting	12 participants completed questionnaires assessing attitudes towards 3D printed food, and tasted snack bars printed using a Fuse Deposition Modelling printer	3D printed snack bars made of cookie dough with filling, prepared similar to commonly consumed snack by soldiers following training, one benchmark snack	Participant attitude towards 3D printed foods improved following repeated tasting sessions, although mean liking scores for the 3D printed snacks were lower than those of the benchmark snack	[52]
Develop a fruit-based food formula nutritionally designed for children to build 3D snacks	3D Printer mod. Delta 2040 equipped with the Clay extruder kit 2.00.	3D printed snack consisting of bananas, dried mushrooms, canned white beans, lemon juice, non-fat milk, pectin powder, ascorbic acid	Fruit-based formula estimated to contain 5–10% of required energy, calcium, iron and vitamin D for children ages 3–10 Flow of material affected the dimensional and microstructural properties of the product	[55]
Investigate the printability of fiber-enriched foods with acceptable organoleptic characteristics, using button mushrooms	20 semi-trained panellists tasted snacks made using an in-house built CARK delta-type extrusion-based 3D food printer	Snacks prepared with freeze-dried white button mushrooms, wheat flour, potassium metabisulphite, sodium chloride, calcium chloride, post-processed as sweet or spicy snacks	Panellists liked the product overall and preferred the spicy products over sweet products. Freeze-dried powder was not printable but addition of wheat flour improved printability; optimal printing speed, nozzle size and flow rate were determined	[59]
Assess the printability of and design healthy food structures high in fiber, protein and low in fat or sugar	VTT micron scale dispensing environment based on nScrypt technology	Snacks made with starch, milk powder, cellulose nanofiber, rye bran, oat protein and fava bean protein concentrates	Optimal parameters were identified for best printing results, such as composition of pastes; higher yield stress resulted in acceptable shape stability	[60]
Investigate the printability of cereal-based food structures with probiotics	ByFlow 3D food printer	Wheat dough with different flour/water ratios, optimized for printing, supplemented with Lactobacillus plantarum WCFS1 (LP)	Following baking, 3D printed structure was maintained and acceptable viability of the probiotics was obtained; suggestions for further modifications to enhance probiotic survival were made	[56]

## Data Availability

The datasets generated for this study are available on request to the corresponding author.

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
