# Peer review of "Three-Dimensional Printing of Foods: A Critical Review of the Present State in Healthcare Applications, and Potential Risks and Benefits"

_foods, 2023, doi:10.3390/foods12173287_

Round 1

Reviewer 1 Report

The review provided a comprehensive summary on the background, application, and challenges of food 3D printing. The review reflected the author’s opinion on food 3D printing, the challenges should be concerned for future researchers or producers. Food safety is the most important aspect. Here are some suggestions:

1.     The title is not very suitable, the author didn’t mention too much about the art of 3D printing. And capitalized words should be unified.

2.     There are no figures or summarized tables in the manuscript. I suggest the authors should add a table to summarize the past literature on various 3D printing food in the Promises parts.

3.     In the Current Applications part, I suggest the authors add some figures of the 3D printing machine mentioned in the text.

4.     In the Challenges part, I suggest the authors give a schematic illustration of the main challenges, where are the challenges from, and how to solve them.

5.     The overall manuscript reflects the deep insight into the field. Some figures and tables will make the manuscript easier to read. 

Author Response

Reviewer 1

The review provided a comprehensive summary on the background, application, and challenges of food 3D printing. The review reflected the author’s opinion on food 3D printing, the challenges should be concerned for future researchers or producers. Food safety is the most important aspect. Here are some suggestions:

Response: We thank the reviewer for the helpful suggestions, please see below our responses.

  1. The title is not very suitable, the author didn’t mention too much about the art of 3D printing. And capitalized words should be unified.

Response: The reviewer makes a valid point. The title was reworded to ”3D Printing Foods: A Critical Review of the Present State in Healthcare Applications, and Potential Risks and Benefits”

  1. There are no figures or summarized tables in the manuscript. I suggest the authors should add a table to summarize the past literature on various 3D printing food in the Promises parts.

Response: We thank the reviewer for this suggestion, and have added a table summarizing relevant research into advances in 3D food printing in the Promises section, titled Table 1: Overview of studies investigating the printability of novel foods, consumer acceptability and/or applicability to potential functional products.

  1. In the Current Applications part, I suggest the authors add some figures of the 3D printing machine mentioned in the text.

Response: We have added a figure describing different 3D printing machines titled Figure 1: Schematics of some examples of commercially available 3D food printers.

  1. In the Challenges part, I suggest the authors give a schematic illustration of the main challenges, where are the challenges from, and how to solve them.

Response: We have added a schematic outlining the main challenges faced by 3D food printing technology, as suggested, titled Figure 2: Challenges that need to be overcome to allow widespread applications of 3D food printing technology in the healthcare and other sectors.

  1. The overall manuscript reflects the deep insight into the field. Some figures and tables will make the manuscript easier to read.

Response: We appreciate the positive and constructive comments from the reviewer that have aided in improving the clarity and presentation of the review.

Reviewer 2 Report

Dear authors,

            The manuscript is a simple narrative one and the flow is appropriate, however, it needs further improvements in order to be published. I recommend ameliorating the structure, adding more information regarding the innovative point of your study, and widely discussing the factors involved. Tables and figures would be highly recommended. The potential risks are briefly discussed.  Please find below some comments that need to be addressed, point by point.

Comments:

Line 14: the description “useful for dysphagia” is quite inappropriate. The dysfunction does not need the food, patients suffering from dysphagia need tailored food to improve their health condition. Please reformulate.

Lines 99-101: Please exemplify. How can 3D printing use microencapsulation technology? What are the benefits? I suggest you elaborate on this idea. Moreover, connect this concept with the next one from lines 101-103.

The article has a nice flow but is a simple one. The innovation element is missing, and I feel like you remained on the surface of the subject and did not dive into it. You have indeed highlighted important “Pitfalls” which I suggest you improve with more scientific information.

Lines 396-411 present valuable information which I suggest you improve by describing in more depth the mechanism in the gastrointestinal tract and the factors affecting the amounts of energy provided by 3D-printed foods.

Also, I recommend diving into the relationship between Gut Microbiota, Circadian Rhythms and ingested 3D-printed foods.

How does Gut Microbiota influence the assimilation of nutrients found in 3D-printed foods?

How do circadian rhythms influence the assimilation of nutrients found in 3D-printed foods?

The relationship between dysbiosis and the bioavailability/bioaccessibility of nutrients might represent an exciting subject for your readers.

I am not a native English speaker, therefore I would prefer to  omit any comments related to the quality of English Lenguage. 

Author Response

Reviewer 2

The manuscript is a simple narrative one and the flow is appropriate, however, it needs further improvements in order to be published. I recommend ameliorating the structure, adding more information regarding the innovative point of your study, and widely discussing the factors involved. Tables and figures would be highly recommended. The potential risks are briefly discussed. Please find below some comments that need to be addressed, point by point.

Response: We thank the reviewer for these valuable comments. We have added subheadings to the different sections of the review to improve readability and structure of the text, and have added a table and two figures:

Table 1: Overview of studies investigating the printability of novel foods, consumer acceptability and/or applicability to potential functional products.

Figure 1: Schematics of some examples of commercially available 3D food printers.

Figure 2: Challenges that need to be overcome to allow widespread applications of 3D food printing technology in the healthcare and other sectors.

In terms of the innovative aspects of our review, most reviews to date regarding 3D food printing have focused on the technological advances and challenges, and some have touched upon regulatory issues that should be addressed to allow the widespread use of this technology. Our review focuses on the health and nutritional considerations with respect to applications of this technology, and although some of these issues have touched upon these concepts, the present manuscript encompasses a wide range of aspects regarding health effects and applications, but also provides a critical perspective regarding potential challenges and nutritional/health risks that must be considered.

Line 14: the description “useful for dysphagia” is quite inappropriate. The dysfunction does not need the food, patients suffering from dysphagia need tailored food to improve their health condition. Please reformulate.

Response: We thank the reviewer for this valid point. We have rephrased the sentence as follows: “... create uniquely textured foods tailored to patients with dysphagia”.

Lines 99-101: Please exemplify. How can 3D printing use microencapsulation technology? What are the benefits? I suggest you elaborate on this idea. Moreover, connect this concept with the next one from lines 101-103.

Response: We have added the following text to this section in order to address this point: “Among the benefits, microencapsulation allows the incorporation of bioactive hydrophobic substances such as tocopherols and polyphenols with low water solubility in food 3D printing technology, which can circumvent issues such as poor bioavailability and sensitivities to degradation due to temperature, pH and oxidation.”

The article has a nice flow but is a simple one. The innovation element is missing, and I feel like you remained on the surface of the subject and did not dive into it. You have indeed highlighted important “Pitfalls” which I suggest you improve with more scientific information.

Lines 396-411 present valuable information which I suggest you improve by describing in more depth the mechanism in the gastrointestinal tract and the factors affecting the amounts of energy provided by 3D-printed foods.

Response: We have added more information in the “Pitfalls” section that includes further cited scientific studies:

In “Health risks of ultraprocessed foods”:

“For example, a two week in-patient intervention trial showed that an ultra-processed diet resulted in weight gain and enhanced caloric intake of approximately 500 kcal/day. In contrast, an unprocessed diet to the same subjects containing similar amounts of calories, macronutrients, fiber, sugars, sodium and fiber resulted in spontaneous weight loss [101]. A variety of mechanisms of action have been postulated for such findings, which includes that over-consumption of calories is reinforced from the intake of hyper-palatable, ultra-processed foods high in sugar and fat [102].

In “Loss of food matrix”:

“Loss of the natural food matrix has been indicated to lead to altered patterns of intestinal digestion and absorption of food components. Nutrients from ultra-processed foods may be absorbed in more proximal regions of the gut, which can lead to differing patterns of gut-brain signals from those generated from the same nutrients from whole food intake [101]. Signals affecting food intake and digestion, energy balance and body weight come from enteroendocrine hormones (i.e., ghrelin, cholecystokinin, glucagon like peptide 1, peptide YY, insulin like peptide 5), vagal afferent nerves innervating the gastrointestinal tract (the neuronal component of the gut-brain axis) and microbiota-derived metabolites (i.e., short-chain fatty acids, amino acids, bile acids, γ-aminobutyric acid) [102].”

Also, I recommend diving into the relationship between the Gut Microbiota, Circadian Rhythms and ingested 3D-printed foods.

How does the Gut Microbiota influence the assimilation of nutrients found in 3D-printed foods?

How do circadian rhythms influence the assimilation of nutrients found in 3D-printed foods?

The relationship between dysbiosis and the bioavailability/bioaccessibility of nutrients might represent an exciting subject for your readers.

Response: As 3D food printing is a recent field of study, there is no research to date regarding the effects of gut microbiota or circadian rhythms on nutrient assimilation, or on the relationship between dysbiosis and bioavailability/bioaccessibility of nutrients in the context of 3D-printed foods. Nonetheless, we have added the following information at the end of the section “Loss of food matrix”: “For example, the obesogenic effects of food additives such as emulsifying agents demonstrated in animal models has been related to altered gut microbiota profiles as seen by germ-free mouse and fecal transplant studies [108].

Reviewer 3 Report

The manuscript is well written and will be a good read to extend the existing understanding of the subject. the manuscript would help with addition of proper subheadings in-between sections to allow continuity and integration of information.

None

Author Response

The manuscript is well written and will be a good read to extend the existing understanding of the subject. The manuscript would help with addition of proper subheadings in-between sections to allow continuity and integration of information.

Response: We appreciate the positive feedback on our manuscript and the constructive comment. We have added sub-headings in-between sections for better readability as suggested.

Reviewer 4 Report

Undertaking a review of the related literature is an important part of any discipline. It helps to maps and assesses the existing knowledge and gaps on specific issues which will further develop the knowledge base. Literature reviews need to adopt a replicable, scientific, and transparent procedures. It helps to collect all related publications and documents that fit pre-defined inclusion criteria to answer a specific research question. It uses unambiguous and systematic procedures to minimize the occurrence of bias during searching, identification, appraisal, synthesis, analysis, and summary of studies. When the procedure is done properly and has the minimal error, the study can provide reliable findings and reliable conclusion that could help decision-makers and scientific practitioners to act accordingly. Well done procedure for the systematic literature review process is essential and it ensures that the work is carefully planned before the actual review work starts.

Therefore, the systematic reviews (like this one) must have a methods section. This section enables motivated researchers to repeat the review. If for any reason, the authors would like to avoid a separate materials and methods section, then they should include some information about applied methods at the end of the introduction. The information should contain data sources (e.g., bibliographic databases), search terms and search strategies, selection criteria (inclusion/exclusion of studies), the number of studies screened, and the number of studies included etc.

Unfortunately, this review does not contain a methodology section and that might be the reason why it has not included most recent manuscripts relevant to this topic like 10.1016/j.cofs.2021.03.015

Additionally, abstracts of scientific papers are sometimes poorly written, often lack important information. Although some journals still publish abstracts that are written as free-flowing paragraphs, most journals require abstracts to contain the usual sections like Background, Methods, Results, and Conclusions. The abstract of this manuscript acts as an introduction to the manuscript and misses to provide the readers with important info regarding most important findings and conclusions. Therefore, it needs to be completely rewritten.

Minor editing of English language required.

Author Response

Undertaking a review of the related literature is an important part of any discipline. It helps to maps and assesses the existing knowledge and gaps on specific issues which will further develop the knowledge base. Literature reviews need to adopt a replicable, scientific, and transparent procedures. It helps to collect all related publications and documents that fit pre-defined inclusion criteria to answer a specific research question. It uses unambiguous and systematic procedures to minimize the occurrence of bias during searching, identification, appraisal, synthesis, analysis, and summary of studies. When the procedure is done properly and has the minimal error, the study can provide reliable findings and reliable conclusion that could help decision-makers and scientific practitioners to act accordingly. Well done procedure for the systematic literature review process is essential and it ensures that the work is carefully planned before the actual review work starts.

Therefore, the systematic reviews (like this one) must have a methods section. This section enables motivated researchers to repeat the review. If for any reason, the authors would like to avoid a separate materials and methods section, then they should include some information about applied methods at the end of the introduction. The information should contain data sources (e.g., bibliographic databases), search terms and search strategies, selection criteria (inclusion/exclusion of studies), the number of studies screened, and the number of studies included etc.

Unfortunately, this review does not contain a methodology section and that might be the reason why it has not included most recent manuscripts relevant to this topic like 10.1016/j.cofs.2021.03.015

Response: The reviewer raises an interesting issue. The review that was conducted can be characterized as a critical review, i.e.,  to critically evaluate and provide conceptual innovations regarding the current state of knowledge based on the most significant issues in the field. This latter approach differs from a systematic review where there are time and scope constraints. Therefore, a methodology section is not usually warranted for a narrative review such as this one.

Additionally, abstracts of scientific papers are sometimes poorly written, often lack important information. Although some journals still publish abstracts that are written as free-flowing paragraphs, most journals require abstracts to contain the usual sections like Background, Methods, Results, and Conclusions. The abstract of this manuscript acts as an introduction to the manuscript and misses to provide the readers with important info regarding most important findings and conclusions. Therefore, it needs to be completely rewritten.

Response: The reviewer makes a good point, as the abstracts for systematic reviews must present Background, Methods, Results and Conclusion sections. However, this review is a narrative review that provides a critical perspective regarding potential challenges and nutritional/health risks that must be considered regarding 3D food printing. Therefore, our abstract has been designed as a summary of the main points discussed throughout the review, such as the potential applications of 3D printed foods (applications for dysphagia, sustainability, reduction of food waste), the main challenges faced by 3D printing (nutritional quality, manufacturing issues), the potential nutritional impact (nutrient density, bioaccessibility/bioavailability, loss of food matrix), consumer acceptability, food safety, and the possible adverse health effects due to overconsumption of 3D printed foods.

Round 2

Reviewer 2 Report

The authors improved the manuscript by adding representative figures and tables, as well as explanatory and additional information to the text. 

The modified version of the manuscript is well-structured and comprehensible, therefore it can be published by the journal of Foods. 

Author Response

We thank the reviewer for their helpful comments.